# Aquatic Environment Exposure and Toxicity of Engineered Nanomaterials Released from Nano-Enabled Products: Current Status and Data Needs

**DOI:** 10.3390/nano11112868

**Published:** 2021-10-27

**Authors:** Mbuyiselwa Shadrack Moloi, Raisibe Florence Lehutso, Mariana Erasmus, Paul Johan Oberholster, Melusi Thwala

**Affiliations:** 1Centre for Environmental Management, University of the Free State, Bloemfontein 9031, South Africa; 2012164071@ufs4life.ac.za (M.S.M.); OberholsterPJ@ufs.ac.za (P.J.O.); 2Water Centre, Council for Scientific and Industrial Research, Pretoria 0001, South Africa; flehutso@csir.co.za; 3Centre for Mineral Biogeochemistry, University of the Free State, Bloemfontein 9031, South Africa; ErasM@ufs.ac.za

**Keywords:** product released nanomaterials, nano-enabled product, risk assessment, ecotoxicology of PR–ENMs

## Abstract

Rapid commercialisation of nano-enabled products (NEPs) elevates the potential environmental release of engineered nanomaterials (ENMs) along the product life cycle. The current review examined the state of the art literature on aquatic environment exposure and ecotoxicity of product released (PR) engineered nanomaterials (PR–ENMs). Additionally, the data obtained were applied to estimate the risk posed by PR–ENMs to various trophic levels of aquatic biota as a means of identifying priority NEPs cases that may require attention with regards to examining environmental implications. Overall, the PR–ENMs are predominantly associated with the matrix of the respective NEPs, a factor that often hinders proper isolation of nano-driven toxicity effects. Nevertheless, some studies have attributed the toxicity basis of observed adverse effects to a combination of the released ions, ENMs and other components of NEPs. Notwithstanding the limitation of current ecotoxicology data limitations, the risk estimated herein points to an elevated risk towards fish arising from fabrics’ PR–nAg, and the considerable potential effects from sunscreens’ PR–nZnO and PR–nTiO_2_ to algae, echinoderms, and crustaceans (PR–nZnO), whereas PR–nTiO_2_ poses no significant risk to echinoderms. Considering that the current data limitations will not be overcome immediately, we recommend the careful application of similar risk estimation to isolate/prioritise cases of NEPs for detailed characterisation of ENMs’ release and effects in aquatic environments.

## 1. Introduction

The advancement of nanotechnology has increased the frequency of engineered nanomaterials’ (ENMs) incorporation into products in pursuit of their superior properties to enhance product formulations. For instance, ENMs are favoured because they exhibit relatively increased surface area and pore volume [1], which facilitates enhanced adsorption, ion exchange and increased reactivity [1,2]. Product formulations containing ENMs are called nano-enabled products (NEPs), and their global market is rapidly increasing being forecast to be worth USD 125 billion in 2024, from *ca* USD 32.9 billion in 2016 [3]. The number of NEPs listed in various global inventories has also increased year on year, from 54 in 2005 [4] to above 5000 in 2020 [5]. In approximately over a decade, nanotechnology has advanced from the research and development phase to daily use in products [5].

The global NEPs’ markets are currently dominated by health and fitness products, mainly active wear, sunscreens, cosmetics and sporting goods [4,6,7,8]. In health and fitness products, the ENMs are predominantly surface-bound or suspended in liquid; a character that increases their potential release into water environments (medium to high environmental exposure potential) [6,7,9,10].

The ENMs in NEPs are commonly not permanently fixed in the product matrix and can be released into the environment during product use and the end of life stages. The emission of ENMs from NEPs into the environment, nanopollution, is expected to be proportional to the rising commercialisation of NEPs. The ENMs released from NEPs are referred to as product released (PR) engineered nanomaterials (PR–ENMs) to distinguish them from pristine counterparts (bare or not incorporated in products).

Data on the environmental risks associated with PR–ENMs are limited as their hazard, exposure dynamics and toxicity effects are not well characterised [7,9,10,11,12]. Partly, this can be attributed to the limited suitability of current analytical techniques to examine ENMs in complex media as well as the fairly recent emergence of the focus on examining ENMs’ environmental risks arising from NEPs. For instance, the NEP inventory focusing on establishing NEPs’ market penetration was initiated in 2005 [4], while those with an interest on ENMs’ environmental exposure potential were only introduced in 2012 [6], 2015 [8] and 2019 [7], respectively, from Europe, Singapore and South Africa. Similarly, studies that experimentally investigated the environmental exposure potential from NEPs such as paints, sunscreens and textiles emerged around 2008–2010 [13,14,15]. While challenges concerning risk determination approaches currently persist, various guides have been developed pertaining to ENMs’ characterisation requirements and strategies to mitigate potential risks, for instance, by the Food and Drug Administration (FDA) in the USA [16,17], European Commission [18] and the Organisation for Economic Development and Co-operation (OECD) [19].

The examination of PR–ENMs’ environmental exposure and effects are hindered predominantly by the low concentrations (ranging in the lower ppb) that are released into the environment, thus limiting analytical characterisation (exposure) and hazard assessment [9,20]. For instance, as little as 0.007 to 0.5% of PR–ENMs (nAg and nTiO_2_) was released from textile and paint products’ NEPs [20,21]. From sunscreens, 0.16–1.16 µg/L nTiO_2_ was released into water environments [22]. Overall, the low release amounts and limited suitability of current analytical tools for “nano” characterisation in complex media have been raised as priority challenges [10,23,24,25] and various detailed reviews have focused on this matter [26,27,28,29,30,31].

Concerning the exposure of aquatic systems, wastewater treatment plants (WWTPs) have been identified as sinks and secondary sources of ENMs into water resources (Figure 1) and, thus, should be considered in PR–ENMs’ environmental exposure and risk assessment. For instance, nZnO and nTiO_2_ concentrations in WWTPs have been correlated with the usage of NEPs [32]. Considerable amounts of the PR–ENMs can be retained in sludge, for instance, with a sorption density of 3.49 g/kg mixed liquid suspended solid (MLSS) for nZnO and 4.67 g/kg MLSS for nTiO_2_ [33], while sludge application as a fertiliser in agricultural fields can emit PR–ENMs into aquatic bodies during runoff events [34].

The current paper reviews the state of the art literature concerning the characterisation of release and toxicity effects of PR–ENMs in aquatic ecosystems and their respective risk to guide the identification of product emissions that could pose notable unwanted consequences. Specifically, the risk estimation was meant to direct research and regulation attention to products that need prioritization for ENMs’ aquatic exposure and hazard characterisation. The review identifies key gaps and outlines priority research needs on the environmental exposure and hazardous effects of PR–ENMs in water systems. Specific to the release and toxicity of PR–ENMs in real aquatic environments, only a few studies exist [35,36,37]. The current review is the first to consolidate the release and effects data for risk estimation as the handful of previous reviews have mainly presented the data on PR–ENMs [12,20,27,38,39,40,41,42,43,44,45].

The data reviewed herein were obtained from peer-reviewed articles in Google Scholar and Science Direct databases. Articles were limited to the 2008–2021 period and filtered using the following keywords: product released nanomaterials, nano-enabled products, engineered nanomaterial released from products and toxicity effects of engineered nanomaterial released from nano-enabled products. A total of 322 peer-reviewed articles were obtained at the first tier, but 181 were discarded from further analysis as they did not fit the scope of this review; thus, 142 articled were reviewed.

## 2. ENMs Identified to Be Commonly Used in NEPs That Exhibit Medium to High Environmental Exposure Potential

Generally, ENMs are incorporated in NEPs to enhance specific product properties based on the character of the ENMs being applied [46,47,48,49]. This section reviews the beneficial properties of ENMs (titanium dioxide (nTiO_2_), zinc oxide nanoparticles (nZnO), silicon dioxide nanoparticles (nSiO_2_) and silver (nAg) that make them attractive for incorporation in NEPs. The ENMs’ sample was selected because of the high production rates (Table 1), wide distribution across product categories and high usage in NEPs.

Twenty–five percent (25%) of all NEPs in the consumer market are reported to be incorporated with nAg, nTiO_2_ or nSiO_2_ in [52]. Moeta et al. [7] and Zhang et al. [8] also reported that from the 264 and 1432 NEPs surveyed, respectively, nTiO_2_, nSiO_2_, nAg and nZnO were the most common ENMs in products found in South Africa (74%) and Singapore (>80%). Similarly, these ENMs were found in most of the NEPs found in Europe (Figure 2). In all inventories [4,7,8,53,54], these ENMs were predominant in NEPs categorised as having medium to high environmental release potential, suggesting that they were likely to be emitted to the environment with relative ease.

### 2.1. Titanium Dioxide Nanoparticles (nTiO_2_)

nTiO_2_ is one of the most highly produced ENMs globally (Table 1) as a white pigment applied in many NEPs [47]. Additionally, nTiO_2_ is favoured for its brightness, resistance to discolouration, high refractive index and broad ultraviolet (UV) spectrum that allows for maximum blockage and protection from UV radiation [55,56]. nTiO_2_ is used in pharmaceuticals as photosensitisers in photodynamic therapy [57], in cosmetics as sun filters in day creams and foundations [58], and in toothpaste as a white pigment [59,60]. The high usage of nTiO_2_ in cosmetics and fabrics for its UV filtration [55,56,61], high refractive index and as a colouring agent in paints [62,63], as well as its self-cleaning property in textiles and paints [62,64] has led to the continuously rising production over the years.

nTiO_2_ is produced in three crystalline forms: anatase, rutile and brookite; the former two are the most commonly found in products [59,65]. Anatase and rutile are predominantly used in NEPs due to their high energy absorbing properties [66,67]. Rutile is a more stable form of nTiO_2_, while anatase is metastable and transforms into the rutile phase at elevated temperatures [68]. From a sample of six sunscreens, rutile nTiO_2_ was present in three sunscreens, while the combination of rutile and anatase forms of nTiO_2_ was in one sunscreen [10,22,69]. Overall, most studies demonstrate that rutile nTiO_2_ is the most widely used form [10,58,59,70,71,72,73]. Rutile nTiO_2_ has also been reported in paints [10] and cotton fabrics [61]. Although both anatase and rutile nTiO_2_ are widely applied in products, rutile appears to be the most preferred derivative, probably due to its relatively superior UV absorbance, lower photoreactivity and higher photocatalysis than anatase [61,72,74].

In addition to the form of nTiO_2_, its size, shape and surface coating characteristics have also been reported. Bairi et al. [72] and Lu et al. [69] have confirmed smaller than 100 nm nTiO_2_ in products. In sunscreens, nTiO_2_ has also been reported to be in the size range of 8–34 nm (irregularly/angularly shaped) [7,10,75,76,77,78] and 44.4–96.3 nm (needle shaped) [10,22,79,80,81,82].

In cotton fabrics, the size of nTiO_2_ has been found to be 20–80 nm [61,83,84]. In paints, an nTiO_2_ size of 100–450 nm has been reported [10,21] being either angular or needle shaped [10,21,72,85,86,87]. Due to the photocatalytic property of nTiO_2_, before its incorporation in NEPs, the surface is commonly coated with agents such as silicon dioxide (SiO_2_), aluminium oxide (Al_2_O_3_) and aluminium hydroxide (Al(OH)_3_) [10,22,70,77,88,89,90]. The coating increases the photostability of the ENMs, reduces photoreactivity and the formation of reactive oxygen species and improves dispersion in the product matrix [69,90,91,92,93].

### 2.2. Zinc Oxide Nanoparticles (nZnO)

nZnO is the third highest produced ENM type worldwide because of its wide application as an active ingredient in sunscreens and cosmetic products [69,72,94,95] in textiles, wound healing, anti-haemorrhoids, antibacterial agents and eczema medical treatments [44,94]. The myriad applications of nZnO can be attributed to their high chemical stability and electrochemical coupling, paramagnetic nature, a broad range of radiation absorption and high photostability [44]. nZnO has also been reported to have the broadest UV protection and is thus utilised as an active ingredient for most of the UV-blocking products available in the markets [95,96,97].

nZnO is produced in two crystalline forms, namely wurtzite and zincite [71,98]. The hexagonal wurtzite phase of nZnO is more stable and is thus widely incorporated in NEPs [71]. Lewicka et al. [99] found that 4 out of 11 sunscreens contained hexagonal wurtzite phase nZnO. Elsewhere, the presence of crystal wurtzite nZnO was similarly reported [72]. However, Lehutso et al. [10], reported the presence of zincite phase nZnO in a sunscreen that contained binary nTiO_2_ + nZnO [10]. Regarding shape and size, nZnO was reported to be angularly shaped (20–100 nm) [10,69,99,100] and rod shaped [10,69,79,101].

### 2.3. Silver Nanoparticles (nAg)

Silver has long been known for its antimicrobial properties [102] and nAg has, accordingly, been widely utilised in NEPs, largely for disinfecting purposes [103,104,105]. nAg has gained popularity in recent years as the preferred silver derivative for use in NEPs because of its increased bioavailability, high specific surface area and a high fraction of surface atoms [106]. The maximised surface area of nAg means the highest possible effect per unit silver compared to bulk forms [107]. The antimicrobial nature and subsequent extensive application of nAg are mainly influenced by their broad-spectrum antibacterial and antifungal activity [108].

Though there is still an ongoing debate on the antibacterial mode of action of nAg [106], it has been observed that bulk Ag can precipitate bacterial cellular proteins and block the respiratory chain system, while nAg, because of its small size, attaches itself to the bacterial cell membrane and penetrates internally where it generates oxidative stress, thus eliminating the bacteria [109]. In essence, bulk Ag toxicity depends on the release of ionic Ag+, whereas nAg effectiveness is additionally linked to the release of free radicals (i.e., oxidative stress induction) even under limited dissolution [106,110,111,112,113,114]. The large surface area of nAg also facilitates its exposure to O_2_ and enhances the dissolution and sorption of ionic species [104]. However, the nAg mode of toxicity for some biota has not been described, although dissolution-linked toxicity has been reported in aquatic higher plants [115,116]. Thus far, nAg has been used in a variety of commercial NEPs such as soaps, pastes, toothbrushes and textiles [25,103,117,118,119,120], paints [23,121], cleaning products [122] and washing machines [123].

The physicochemical properties of nAg in textiles have been reported to be in the range of ~10–100 nm, with near-spherical, spherical and irregular shapes [10,24,124,125,126]. In one report, the zeta potential of nAg was reported to be negative (−16.3 ± 0.5 mV) [124]. In paints, nAg was determined to be <15–100 nm [23,127]. nAg in household surface cleaners was reported to be in the range of <10–101 nm [128,129,130] in quasi-spherical, and spherical shapes [128,129]. The nAg zeta potential range in household surface cleaners has been reported at −38.6 to −54 mV [130].

### 2.4. Silicon Dioxide Nanoparticles (nSiO_2_)

The application of nSiO_2_ in NEPs is rapidly growing with an estimated 198-kilo tons consumed in 2015 and expected to rise to 786-kilo tons by 2022 [52]. The utilisation of nSiO_2_ in NEPs is predominantly for properties including anti-caking, anti-baking, increased absorption and self-cleaning, transparency and abrasion resistance [7,131]. nSiO_2_ is incorporated as an auxiliary material in paper and textile manufacturing [132], food additives [132,133], paints [134], as a pharmaceutical skin treatment for insect bites [135], glass cleaning products [135] and cosmetics (eye creams and hair care products) [7,10].

The relative ease of nSiO_2′_s surface modification is one of the practical properties that promotes application in a wide spectrum of NEPs’ categories [7,132,134,136]. Furthermore, nSiO_2_ has hydrophilic–hydrophobic characteristics [2] and also exhibits high stability and compatibility with other polymers and molecules [136]. Due to their transparency, nSiO_2_ are also incorporated as additives in printer toners, varnishes, paint and food [137] The extensive application of nSiO_2_ in NEPs suggests notable environmental release potential.

While nSiO_2_ has a relatively wide utilisation in NEPs, their physicochemical properties remain undetermined. In the study of Lehutso et al. [10], an NEP labelled as containing nSiO_2_ was found to not contain it [10]. Currently, the data on the physicochemical properties of nSiO_2_ in NEPs remain poorly known, and the status is likely due to its generally low hazard potential; hence, the limited interest in examining its environmental consequences and the priority being afforded to counterparts that are suspected or known to exhibit negative consequences.

## 3. Release of ENMS from NEPs

The ENMs in NEPs may not be permanently fixed into the products matrix, and along the product life cycle, ENMs can be released into the environment [25,40,91,101,120,138]. However, data on the characteristics and level of release extent of product released ENMs (PR–ENMs) are currently limited.

By 2017, 96 studies had been undertaken to examine the release of ENMs from NEPs into the environment [20], although, of those, only 36 contained relevant quantitative data, indicating significant gaps in ENMs’ environmental exposure dynamics. Koivisto et al. [20] reviewed 36 studies covering all environmental recipients and focused on developing a library of released concentrations [20]. Since then, additional studies have been published and are included in this review.

Herein, we only reviewed PR–ENMs’ environmental exposure data specific to aquatic environments, reporting on release methods and physicochemical properties observed. For aquatic exposure, thus far, ENMs’ release has been investigated from sunscreens, personal care products, paints, textiles/clothing, washing machine, baby products and toothbrushes (Table 2).

### 3.1. Sunscreens

Sunscreens are commonly formulated with nTiO_2_ and nZnO, and several studies have examined the release of the ENMs using various methods. In one of the early studies [70], nTiO_2_ was released from different sunscreens by agitation under light and dark conditions over 48 h. Total Ti released from the sunscreens ranged from 19–32 wt% (dark condition) and 22–38 wt% light conditions. The PR–nTiO_2_ were needle-like shaped, averaging 50 × 10 nm (width × length) in size, rutile form and negatively charged. The PR–nTiO_2_ also aggregated up to 50–200 nm [59]. Nthwane et al. [22] mimicked bathing conditions with tap and deionised (DI) water to examine nTiO_2_ release from three sunscreens [22]. The total released Ti was, respectively, 1.16 and 0.7 µg/L in DI water and tap water. The PR–nTiO_2_ were needle-like in morphology and sized 32.1–102.8 nm and 77.6–139.6 nm in DI water and tap water, respectively [22].

Wong et al. [100] examined the release of PR–nZnO from sunscreen use in seawater where 0.5 g of sunscreen was applied to the hands of human volunteers for 20 min and washed off through soaking in artificial seawater for another 20 min [100]. The PR–ENMs release rate was in the order of 8–72% [100]. No further characterisation was undertaken after release.

Elsewhere, nTiO_2_ and nZnO were released by applying sunscreen on pigskins and washing them off by stirring in tap water [76]. From the liquid type sunscreen, ca. 40% of the initial loading of ENMs (nZnO and nTiO_2_) in the sunscreen was released, while the cream type sunscreen released only 20% of the initial ENMs’ (nZnO and nTiO_2_) loading after 120 min of stirring. These findings illustrated that the NEPs’ matrix influences ENMs’ release potential as relatively higher concentrations were released from liquid compared to cream matrix sunscreen. A swimming pool simulation in a similar study further indicated that sunscreens may introduce hydrogen peroxide into the water system [76], free radical species that cause adverse effects on aquatic organisms [139]. A recent study aged a sunscreen through an agitation process over 48 h to characterise the release of ENMs from three sunscreens [140]. Approximately 0.4–8% (*w*/*w*) of the sunscreen’s initial loading of nTiO_2_ and nZnO was released [140]. One sunscreen contained both nTiO_2_ and nZnO; the released nTiO_2_ were elongated, while nZnO was angular in shape. The two other sunscreens released nTiO_2_ that was angularly shaped [140]. The sizes of released ENMs from the mixture sunscreen were 32–36 × 32–40 nm and 7–9 × 66–70 nm (width × length) for nZnO and nTiO_2_, respectively [140]. The sizes of PR–nTiO_2_ were 2–30 × 33–37 and 21–22 × 25–28 nm for sunscreen two and three, respectively. All ENMs released were negatively charged and were suggested to be coated with Al and Si-based surface coatings. Overall, the physicochemical properties of PR–nTiO_2_ and PR–nZnO were relatively similar across the studies, and where differences occurred, were due to factors such as the release method and nature (formulation and matrix) of the NEPs.

**Table 2 nanomaterials-11-02868-t002:** The total concentrations released from different nano-enabled products (NEPs). ENMs = engineered nanomaterials.

NEP	ENMs Type	Concentration/Percentage Released	References
Sunscreen	nTiO_2_	7 × 10^−4^–0.00116 mg/L	[22]
		19–38%	[70]
	nZnO	0.58 mg/L	[100]
	nTiO_2_ + nZnO	20–40%	[76]
		0.4–8%	[140]
Paint	nAg	0.5–20 mg/L	[23]
		3.5 × 10^7^ particles/L	[13]
		1.7–15.7 µg/L	[121]
	nTiO_2_	5 × 10^5^ particles/mL	[127]
		2 × 10^6^–1.2 × 10^7^ µg/m^2^	[74]
		10–30 µg/m^2^	[87]
Textiles	nTiO_2_	0.64–4.7 mg/L	[120]
		0.05 ± 0.02–3.13 ± 1.51 µg/g	[83]
	nAg	0.32–38.5 mg/L	[120]
		<1–100%	[14]
		0.3–377 µg/g	[15]
		18 ± 2–2925 ± 10 mg/kg	[138]
		3.4 ± 0.1–106 ± 10 µg/g	[124]
		15.8–34 µg	[141]
		1 × 10^−3^–5.969 mg/L	[24]
		5.3–6.4 mg/L	[126]
Washing machine	nAg	8.1724 × 10^7^ particles/mL	[123]
Baby products	nAg	1–35%	[142]
Toothbrush	nAg	3.6–6.6 × 10^7^ particles/L (Baby)	[25]
		9.3–20.3 × 10^7^ particles/L (Adult)	[25]

### 3.2. Personal Care Products

Mackevica et al. [83] investigated the release of nAg from adults’ and children’s toothbrushes during use [25]. A baby toothbrush and an adult toothbrush were immersed in tap water and fixed on a rotating rod for 24 h. nAg release from toothbrushes was 9.3–20.3 × 10^7^ particles/L and 3.6–6.6 × 10^7^ particles/L for adults’ and baby’s toothbrushes, respectively. The nAg particle size ranged between 42 and 47 nm and was spherical [25]. Benn et al. [141] reported 100% Ag release from toothbrushes of their initial silver loading.

nAg-containing sanitiser and body cream (incorporated with nAg and binary nAg + nTiO_2_) were investigated for the release of ENMs using different methods [140]. Briefly, the release from the body cream was conducted using an ageing process over 48 h under light and dark conditions, while the release from the hand sanitiser was through agitation over 24 h. The sanitiser’s PR–nAg were near-spherical shaped and sized 10–23 nm, whereas the body cream’s PR-nTiO_2_ was elongated in shape with a size range of 8–9 ± 3 × 60–66 ± 9 nm (width × length) and associated with Si-based coating agents. Similar to the sanitiser, PR–nAg from body cream was near-spherical and had an average size of 12–55 nm. The PR–ENMs of both the sanitiser and body cream were negatively charged: −32.5 ± 2.1 mV for sanitiser and −23.6 ± 1.3 and −22.8 ± 1.2 mV for the body cream in light and dark conditions, respectively [140].

These studies indicate that the introduction of nAg into the environment is possible during use. However, the released concentrations may be lower; contrary to the method used here, the realistic release does not occur continuously over 24 h. Nonetheless, the data presented in these studies provide the starting point for methodology development and optimisation.

### 3.3. Paints

The release of ENMs from paints has been investigated by numerous studies [15,25,79,94,134,140]. The release of nTiO_2_ from paint into surface waters was investigated using painted outdoor façades [13]. One outdoor façade was painted two years prior to the investigation (aged), and the other was painted during the experimental investigation. The runoff from both façades were collected and analysed for nTiO_2_. In the samples from the aged façade, particles in the range of 50–200 nm were detected. Particles with sizes < 100 nm had a concentration of 3.5 × 10^7^ particles/L [13]. The amount of nTiO_2_ was higher in the experimental façade compared to the aged façade.

Kaegi et al. [23] also investigated the release of nAg from paint applied on outdoor building façade panels that were exposed to natural weathering over a year, and the runoff was collected and analysed for nAg [23]. The total Ag released in the runoff was determined to be between 0.5 and 20 mg/L over the exposure period, and there was an average loss of about 30% of the initially applied surface Ag (1.5 mg/m^2^). The PR–nAg was detected as individual particles of <15 nm in size in the runoff. However, the amount of nAg detected in the TEM grids was not sufficient for further characterisation.

Similarly, Künniger et al. [121] used wooden façade panels to evaluate the release of nAg from paint exposed to outdoor conditions for one year [121]. The runoff was collected after every rainfall during the exposure period, and the total Ag concentration released after three months was 0.67 µg/L. During the first three months of exposure, the average Ag concentration released from one of the façade panels ranged between 1 and 21 µg/L. The total concentration of Ag released in the runoff from the second sampling varied between 0.08 and 0.86 µg/L. The total amount of nAg released from the wooden façade panels was 15.7 and 1.7 µg/L for the first and second panel, respectively, both accounting for less than 1% of the nAg in the initial coating. Neither nanoparticle tracking analysis (NTA) nor TEM were suitable for the further characterisation of particulate Ag in the runoff due to very low detection.

Kaegi et al. [127] examined the release of nTiO_2_ from paints on construction and demolition landfill discarded materials [127]. The leachate samples contained spherical nTiO_2_ with sizes of 100–150 nm, and the average concentration of the PR–nTiO_2_ was reported to be 5 × 10^5^ particles/mL. The release of nTiO_2_ from paints was also investigated under actual weathering conditions [85]. The PR–nTiO_2_ was analysed in the collected precipitate made up of mainly snow and rain that had come in contact with painted panels over 10 weeks. Considerable amounts of PR–nTiO_2_ were obtained in the collected precipitate samples; the PR-nTiO_2_ concentration averaged 2–4 × 106 particles/mL in summer and 8.1 × 105 particle/mL to 1.2 × 107 in winter. Overall, 55% of PR–nTiO_2_ was quantified and sized < 60 nm.

Azimzada et al. [74] examined ENMs’ release from the weathering of painted and stained surfaces under different conditions over 23 weeks (11 weeks in fall and 12 weeks in winter) [74]. The painted surfaces released a total of Ti at 10 µg/m^2^ by the end of the fall weathering (11 weeks) and significantly less than 10 µg/m^2^ at the end of the winter weathering (12 weeks). The average size of released PR–nTiO_2_ was <60 nm (15–120 nm). The stained surfaces released over 30 µg/m^2^ PR–nTiO_2_ with a size range of 15–100 nm. Overall, the total release accounted for 5 × 10–5% for painted surfaces and 6% for stained surfaces. The differences in the amount released were affected by the chemistry of the NEPs; the exterior wood stain had smaller particles that were more susceptible to leaching than the pure enamel paint [66]. It is evident that the release of ENMs from paints into the environment occurs, although at low concentrations. The studies on paint, as reported here [25,125,134,140], show a relatively lower total release when compared to the more rapid release studies [70,76]. While the complexity of the paint matrix, especially when it is already dry, influences the ENMs’ release, the environmental conditions to which the test samples were exposed also influence the release.

### 3.4. Clothing/Textile

Several studies have investigated the release of nAg and nTiO_2_ from textiles. nTiO_2_ release was evaluated from six sun-protection textiles during washing [101]. The textiles released a total Ti of <0.7–4.7 mg/L in the washing solution and 0.64 mg/L in the subsequent rinsing solution [101]. A higher amount of total Ti was detected in the first wash and gradually reduced with the latter washes. nTiO_2_ release from synthetic textiles such as wet wipes and microfiber clothes has also been assessed [83], and 0.05 ± 0.02–3.13 ± 1.51 µg/g total Ti was released from the 24-h study. However, only 1% of the initial TiO_2_ content released was in the nanoparticulate form.

Elsewhere, the nAg released from commercial socks in water was investigated using a laboratory washing method [14]. Socks were washed several times and nAg in the release matrix was measured and characterised. The Ag that leached into water ranged from 1.5–650 µg in 500 mL of distilled water. The total silver release ranged from <1% in some socks to 100% of the initial silver loading in other socks. Most of the released Ag was in the ionic form (Ag^+^) [16]. Elemental Ag particles were detected in the release matrix with diameters of 100–500 nm in three of the types of socks. Particles of sizes < 100 nm were of irregular and spherical shapes [14].

nAg release from socks and other nine commercially available fabrics were investigated [15]. Total Ag released was quantified in the release media. Four out of the nine fabrics did not release quantifiable levels of Ag^+^ into the water. The total Ag release from the other five fabrics ranged from 1.3 to 35%, with the amount of Ag released decreasing with subsequent washes [15]. The concentration of released Ag ranged from 0.3 to 377 µg/g [15].

The release of nAg from textiles has also been investigated on eight textile products, with the initial Ag content of 1.5–2925 mg/kg of textile that were washed and rinsed and the wash water analysed for total released Ag [120]. The Ag concentrations in the wash water and rinsing solution were determined to be 0.32–38.5 mg/L and 0.36–22.7 mg/L, respectively, indicating the release of 15–20% [120]. Elsewhere, a standard laboratory method was used to release nAg from five textiles [138]. The textiles released 18 ± 2–2925 ± 10 mg/kg total Ag; the highest release was approximately 80% relative to the initial silver incorporated on/in the textile. In another study, nAg release from four textiles was evaluated by simulating the washing of fabrics [124]. Accordingly, Ag0-coated textile released 106 ± 10 μg/g, which accounted for 26% of the initial silver loading. The nAg textile released 18 ± 3 μg/g, which was 76% of the initial silver loading after four washes. The textile was washed two additional times and released 3.4 ± 0.1 μg/g, 14% of the initial loading and 90% release over six washes. Benn et al. [141] used tap water to wash face masks and shirt fabrics containing nAg for 1 h [141]. The face mask released a total Ag of 15.8 µg/mL into the wash water (<0.01% of the initial loading), while the shirt released 34 µg/mL (2% of the initial loading).

Four textiles containing nAg (cleaning cloth, bodysuit, car sheet, nursing cover) were washed sequentially for 20 consecutive cycles by an agitation method for 30 min per cycle to evaluate nAg release [24]. The total Ag release was 1–5969 µg/L, with the concentrations decreasing with the increasing number of cycles. Over 50% of the Ag content was lost from the bodysuit and the car sheet after 20 cycles, while the cleaning fabric lost only 30% of the initial Ag loading. The released nAg size was 3–5 nm. Gagnon et al. [126] released nAg from socks using actual walking and running conditions [126]. Six people were required to wear the socks without washing, three people walked 8–16 km per week over three weeks, and three others ran for 1 h per week over 3 weeks wearing the socks. After the walk/run periods, the socks were washed in tap water using a liquid detergent and rinsed. The nAg released after the first wash was 6.4 mg/L from walking socks and 5.3 mg/L from running socks [126]. There was no significant difference in the released nAg concentration between the first and the second wash, and the average size of PR–nAg was ca. 50–200 nm [126].

The various studies reviewed herein provide evidence of ENMs released from textiles (nAg and nTiO_2_) into aquatic environments at various stages of use of these fabric products. The extent of ENMs’ release was influenced by the washing cycles, manner of product use, application of wash detergent and initial ENMs’ loading. An aspect that needs further examination is the influence of washing detergents on the released ENMs’ physicochemical characteristics and release extent as there are suggestions of such influence [20] and also to obtain realistic ENMs’ release characteristics. Additionally, the influence of fabric softness is yet to be investigated.

### 3.5. Washing Machine

Farkas et al. [123] investigated the release of nAg from a nano-enabled washing machine (release from machine usage) [123]. The washing machine effluent was analysed for either nAg or ionic silver after fabric wash or use of the silver function. The results showed that nAg concentrations in the effluent were 8.17–8.29 × 107 particles/mL [123]. Based on this data, future studies should also look exclusively at how much of the nAg in the effluent matrix is from the test material and how much is from the washing machine. While the machine used in this study had a specific silver function, the latter point remains.

## 4. Presence of PR–ENMs in Environmental Water Systems

The PR–ENMs enter natural water environments from various sources, predominantly wastewater treatment works; hence, the need to assess the extent and forms of nanopollution in such settings.

Reed et al. [143] examined nTiO_2_ release from sunscreens in surface water arising from recreational activity [143]. Samples collected downstream during the highest recreational activity showed increased total Ti concentrations compared to the background (natural) Ti in the lake (~10 ng/L increase). The total concentration of Ti was 0.4–110 ng/L with size estimated around 79 nm.

Elsewhere, a sampling campaign was carried out in Dutch surface waters to trace ENMs’ release [144]. Water samples collected from 15 points were analysed for PR–nAg, PR–nTiO_2_ and PR-nCeO_2_ using spICP-MS. The total Ag in the river samples was quantified to be 0.3–6.6 ng/L, nAg size averaged 15 nm and the nAg measured environmental concentration (MEC) was determined to be 0.00004–0.619 µg/L. The nAg size was comparable to the waste emissions from textile products that contained nAg; hence, such products were probable sources of nAg accumulation in the two river systems. In the same study, nTiO_2_ was detected in river samples and the total Ti was 0.2–8.1 µg/L, whereas nTiO_2_ size was 250–340 nm. The sources of the determined TiO_2_ were linked to microscale pigments in paints and dyes with the particle size corresponding to the application of nTiO_2_ in food products, toothpaste, sunscreens, cosmetics and drugs. Hence, it can be deduced that these products contributed to the emission of nTiO_2_ in the two river systems. In a different study [144], PR-nTiO_2_ was not detected in the river Dommel.

The presence of nTiO_2_ has also been investigated in a lake at the Old Danube recreational area, Vienna, Austria [145]. Samples were collected from the lake during the bathing and the non-bathing seasons [145]. The PR–nTiO_2_ increased by 40% at the beginning of the bathing season to an average of 9050 ± 3940 particles/mL from 5610 ± 1200 particles/mL during the non-bathing season. The size detection limit was set at 130 nm (lower limit) and PR–nTiO_2_ was confirmed to exist as heteroaggregates; the PR-nTiO_2_ samples were linked to the sunscreen release during bathing [145].

The release of sunscreen ENMs has been examined from bathing activities in the French Mediterranean [146]. The PR–nTiO_2_ and nZnO were detected at elevated amounts during high recreational activity. Total Ti and Zn were, respectively, quantified to be 70–500 and 10–15 µg/L in the top surface water and 10–30 and 3 µg/L in the water column. The particles were recovered as aggregates in the water but were not characterised further.

The confirmation of PR–ENMs in the aquatic environment confirms the occurrence of nanopollution arising from NEPs. However, in most reports, the current nanopollution concentrations are still relatively low but within the range that may affect some sensitive biota (e.g., microbial communities). Furthermore, while the current amounts are still low, the rapidly rising commercialisation of NEPs will probably be accompanied by increasing nanopollution.

Currently, ENMs are not included in routine environmental monitoring programmes due to weak analytical instrumentation and understanding of their exposure and effects; hence they are considered a case of emerging environmental contaminants. In that context, hotspots should be identified and set as priority monitoring sites to provide more clarity on ENMs’ behaviour, fate and effects in aquatic systems, including refining analytical capability.

## 5. Ecotoxicity of PR–ENMs in the Aquatic Environment

In recent years, concerns have been raised about the toxicity of PR–ENMs in the aquatic environment [25,121,147]. This is because there is a rapid increase in NEPs’ global markets and usage [3]. Consequently, the environmental release of PR–ENMs is inevitably rising.

### 5.1. Sunscreen-Released ENMs

A study using a whole product of sunscreen that contained nZnO as an active ingredient evaluated the sunscreen ecotoxicity on copepod *Tigriopus japonicus* (*T. japonicus*) [94]. *T. japonicas* copepods were exposed to three brands of sunscreen in 4 mL of test solutions for over 96 h [100]. One of the sunscreen brands induced the highest toxicity to the copepod species, possibly because of PR–nZnO and released Zn^2+^ [100]. Although significant toxicity was recorded for two of the sunscreen brands, it was observed that the toxicity of nZnO and Zn^2+^ were only partial; other sunscreen components may have contributed to copepod mortality. The median lethal concentration (LC_50_) range was 22.4–230 mg/L depending on the product brand [100].

Similarly, the ecotoxicological response was also evaluated in marine species, namely, *Paracentrotus lividus*, *Phaeodactylum tricornutum*, *Corophium orientalis*, to two types of sunscreens: a chemical-based sunscreen and an organic formulation consisting of metal oxides in their nanoform (nTiO_2_ and nZnO) [148]. The organisms were exposed to the sunscreen in standard saline water and salinity stress at the highest concentrations of 100 µL/L. The effective median concentration (EC_50_) for *P. tricornutum* (chemical sunscreen and nanoformulation), *C. orientalis* (chemical sunscreen), *C. orientalis* (nanoformulation) and *P. lividus* (nanoformulation) was determined to be 96 µL/L under standard salinity after exposure [148]. *P. lividus* exposed to nanoformulation sunscreen exhibited an EC_50_ of 14 µL/L [148]. The nanoformulation sunscreen showed significantly higher toxicity than the chemical-based sunscreen for crustaceans but significantly lower toxicity for algae when compared to the chemical-based sunscreen [148]. Under salinity stress, the EC_50_ for *C. orientalis* was 87 µL/L for the nanoformulation sunscreen and 82 µL/L for the chemical-based sunscreen [148]. The EC_50_ were lower in nanoformulation sunscreen for algae and echinoderms: 9.9 versus 48 µL/L for *P. tricornutum* and 16.9 µL/L versus 71.0 µL/L for nanoformulation and chemical-based sunscreens, respectively, indicating higher toxicity for nanoformulation sunscreen under salinity stress [148]. This effectively means that the physicochemical properties of the exposure medium and sunscreen formulation influenced the observed toxicity. In the study [148], salinity enhanced the toxicity and, therefore, should be taken into account during further investigations that are relevant to coastal and marine environments.

### 5.2. Household Detergent-Released ENMs

The bactericidal effects of PR–nAg in mesosilver used in a household detergent (hot tub cleaner) was investigated using *Pseudoalteromonas aliena* (*P. aliena*), *Cellulophaga fuciola* (*C. fuciola*), *Arthrobacter agilis* (*A. agilis*) and *Streptomyces koyangensis* (*S. koyangensis*) [122]. The exposures (0.062–1.5 mg/L) were developed in 50 mL conical flasks containing 40 mL low nutrient liquid ZM/10 solution at 25 °C in the dark [122]. The PR–nAg inhibited the growth of bacteria at very low concentrations (0.072 mg/L) and compromised their cell viability [122]. Therefore, the release of nAg into the aquatic environment has the potential to induce effects on the ecosystem [149].

### 5.3. Textile-Released ENMs

The toxicity effects of textile PR–nAg have been investigated [124,147]. The toxicity effects of socks’ PR–nAg was assessed on zebrafish embryos (*Danio rerio*) [147]. The *D. rerio* embryos were exposed to undiluted leachate directly from socks and centrifuged leachate over 72 h. Mortality was recorded after 24 h with LC_50_ values at 0.4 and 0.26 mg/L for undiluted leachate and centrifuged leachate, respectively [147]. The nAg-enabled socks’ derived solutions (undiluted sock and spun sock) further induced hatching inhibition and abnormal embryo development after the 72-h exposure [147]. The extensive toxicity in this study was reported to be likely exacerbated by other components in the exposure media other than nAg. Elsewhere, [124] wash water was used to investigate the effects of textile PR–nAg on zebrafish (*Danio rerio*). The wash water from a release study reported earlier in the current paper [147] was diluted to different concentrations, and the zebrafish embryo was statically exposed and assessed at 24-h post-fertilisation [124]. In this investigation, no mortality was recorded and the toxicity of PR–nAg on *D. rerio* embryos could not be validated [124].

### 5.4. Paint-Released ENMs

Künniger et al. [121] evaluated the toxicity of paint PR–nAg where the collected runoff water samples from wooden façades were coated and exposed to the precipitation over one year after rainfall [121]. Toxicity was assessed on algae (*Pseudokirchneriella subcapitata*), bacteria (*Vibrio fischeri*) and daphnia (*Daphnia magna*). The runoff nAg concentrations used for the toxicity assessments were 7.17 µg/L for *V. fischeri* and *P. subcapitata* and 21.08 µg/L for *D. magna* [121]. The toxicological assessment showed that PR–nAg had no significant effects on the tested organisms [121].

The widespread application of ENMs points to continued environmental release and exposure, which may reach levels that are detrimental to the environment. Based on the studies reviewed in this paper, it is evident that several factors need to be considered in assessing the toxicity of PR–ENMs. This includes physicochemical transformations of released ENMs to identify the hazard basis. The co-occurrence of ENMs with product components and wastewater parameters (including binary ENMs) is another key aspect as, currently, the observed toxicity cannot be attributed to just ENMs. Thus far, the synergistic effects of nZnO and nTiO_2_ have been hinted at in this study. However, the impact of binary PR–ENMs is not yet clear and more data still need to be generated on this aspect, although in silico approaches are beginning to provide insights for pristine–ENMs (P–ENMs) [150]. The currently reported aquatic toxicity effects of PR–ENMs discussed in this section are summarised in Table 3.

Most of the reports used mainly P–ENMs, which leaves considerable information paucity on PR–ENMs toxicity. Therefore, more research is required to understand the release, transport, fate and effects of PR–ENMs. It is evident that ENMs used in products such as sunscreens have high environmental exposure potential, and their effects need to be prioritised.

One of the biggest challenges is the low sample mass and concentrations that are extracted or released from NEPs. While realistic, the low concentrations may not induce any significant effect on the organism, as reported by Künniger et al. [121] and Reed et al. [124]. However, the low concentration of ENMs from one-off experiments does not speak to the cumulative effects of bioaccumulation over time. The ENMs have the potential to accumulate in the aquatic systems and their concentrations over time may increase as a result of PR–ENMs’ discharge into these systems, and elongated exposure periods may enhance the ENMs’ hazardous effects, although this has not been reported with PR–ENMs [47,115,151]. While the PR–ENMs may induce effects, the toxicity of other product components is a reality that obscures the current PR–ENM data [124,152].

## 6. PR–ENMs Risk Characterisation Estimation

As illustrated in earlier sections, NEPs can be a source of nanopollution in water resources [11,18,42,152]. Despite persistent challenges pertaining to both exposure and hazard assessments, it has become a necessity to initiate efforts to estimate potential risks, considering the data limitations. To estimate the present risk arising from PR–ENMs in aquatic systems, the current section summarised and utilised available release concentrations and ecotoxicological data to derive the PNECs (predicted no effects concentration). The data from the release studies were used as measured environmental concentration (MEC) without factoring in parameters that can influence exposure potentials, such as product use rate and environmental flows. The risk quotient was calculated using the following
(1)$$RQ=\frac{MEC}{PNEC}$$
where MEC is the measured environmental concentration; MECs of PR–ENMs are reported in release studies (Table 2). The lowest and highest concentrations of PR–ENMs per NEP type represented the least and worst case scenarios, respectively.
(2)$$PNEC=\frac{LC_{50}}{AF}\;\mathrm{or}\;PNEC=\frac{EC_{50}}{AF}$$
where EC_50_ or LC_50_ is the half-maximal concentration or median lethal concentrations in the ecotoxicological studies reviewed, reported earlier in this paper (Table 3). Similar to MECs, the lowest and highest EC_50_ or LC_50_ of PR–ENMs per NEP type represent the worst and least case scenarios, respectively. AF is the assessment factor for any given environmental compartment. The assessment factor of 1000 was applied since all toxicity assessments were examined under acute conditions [18]. The RQ values are interpreted using the following scale: RQ < 1 means no significant risk, RQ of 1–10 means small adverse effects, RQ of 10–100 means potential adverse effects and RQ > 100 means significant adverse effects.

Using Equations (1) and (2) and data sets listed in Table 2 and Table 3, the RQs of PR–ENMs under different scenarios were determined (Table 4).

The RQs were reported for two NEPs (sunscreen and textiles); the limitation was dictated by the availability of data in the reviewed studies; it is indicative that the risk assessment was hampered by the unavailability or shortage of PR–ENMs’ raw data [153]. Modelling studies, therefore, will continue to rely on P–ENMs data [154], which may overestimate or underestimate the risk until the PR–ENMs data gaps (exposure and effects) are addressed.

From Table 4, it can be observed that textile PR–nAg presents the highest RQ for fish (*Danio rerio*) for both the least and worst case scenario (RQ > 100). According to the RQ scale, textile PR–nAg will therefore have significant adverse effects on fish. The least case scenario RQ (800) is in the range of 100–1000 that warrants further testing to verify the risk [18]. The RQ above 1000 in the worst case scenario means that risk reduction measures should be implemented. However, the data analysed in this paper are only enough to give cautious estimates as there is not enough ecotoxicological data for textile nAg to establish the risk.

Sunscreen PR–nTiO_2_ showed the RQ values for crustaceans were <1 for both the worst case and least case scenarios. While there is an increase in the RQ values, the increase is low and remains below 1. For algae, the RQ value was in the range of 10–100 for PR–nZnO and PR–nTiO_2_, but only in the worst case scenario for PR–nTiO_2_. There were similar results for crustaceans. The RQ value here means that there are potential adverse effects from sunscreen PR–ENMs to algae over time. Thus, there needs to be a thorough establishment of the potential risks of these PR–ENMs over time; similar sentiments have been raised previously [34].

The risk estimation, as presented herein, is limited by a few factors, mainly by data paucity regarding the release and effects of PR–ENMs. Ecotoxicological studies are also still lacking, and this means the PNEC values are calculated from minimal available data. For instance, the LC_50_ or EC_50_ values used to calculate PNECs in this study were limited to a few reports of EC_50_ or LC_50_ values (Table 4). Part of that can be attributed to the scarcity of ecotoxicological data that specifically uses PR–ENMs [154]. Secondly, the PR–ENMs’ concentrations were used as MECs, assuming those would be the concentrations that are released into the environment with transformation and dilution not accounted for. While this helps for risk estimation, the actual measured concentrations in the environment will likely be lower and, thus, the risk is overestimated. However, the risk determination is useful to serve as a guide for NEP cases that may require priority attention concerning the generation of ENMs’ exposure and effects. However, the current PR–ENMs’ risk estimation was comparable to other modelling studies [34,154], which also found that cosmetics PR–ENMs are expected to pose a significant risk to the aquatic environment.

## 7. Conclusions, Recommendations and Future Directions

Research on PR–ENMs’ environmental implication has considerably lagged behind nanotechnology advances; however, data on the PR–ENMs of nTiO_2_, nZnO, nAg and nSiO_2_ is starting to grow. The majority of studies investigating the release of ENMs from NEPs commonly report the concentration and/or percentage of ENMs released, but the detailed characterisation of PR–ENMs after release remains rare, partly due to the small sample volume/mass obtainable after conducting release. PR–ENMs’ release into aquatic environments has been reported from sunscreens and other cosmetic products, paints, food products and pharmaceutical products. The heteroaggregation of PR–ENMs with other product components and media abiotic factors is a significant analytical challenge that hinders the “nano” hazard identification. The studies so far have predominantly been on freshwater algae, bacteria and invertebrates (daphnia), while a few exist on marine invertebrates and fish embryos.

Critical questions remain about the environmental risks of PR–ENMs in the face of data challenges related to their exposure and effects. To accelerate data solicitation, we recommend the following:

The development of country/region-specific NEPs inventories to serve as the primary basis for the assessment of PR–ENMs’ environmental release likelihood. Voluntary reporting of some basic properties of ENMs by manufacturers can enrich such a data collection activity.

The examination of all commercially active NEPs is impractical and unnecessary, hence, detailed release and effects assessment of PR–ENMs should be dedicated to NEPs that exhibit medium to high environmental exposure, as defined by the conditions for a specific geographic region. Additionally, assessments should couple laboratory- and modelling-based studies to advance information generation. It is highly encouraged that release studies incorporate realistic conditions that may be at play at various product life cycle stages (prioritising actual products compared to laboratory formulates). However, comparative assessments in a simple medium are essential to eliminate, as much as possible, biotic and abiotic factors that may be introduced, possibly making data evaluation more complex. For instance, well-characterised laboratory constituted aquatic media can serve as valuable control media for assessment that mimics actual product use. Furthermore, standardised methods for ENMs’ release should be developed per product category/type to improve data comparability.

Once PR–ENMs’ exposure likelihood has been obtained and ranked, where environmental monitoring is justified, this should be effected in hot spots for environmental release, for instance, wastewater effluents and discharge points, solid waste dumps and demolition sites for construction materials, etc.

## Figures and Tables

**Figure 1 nanomaterials-11-02868-f001:**
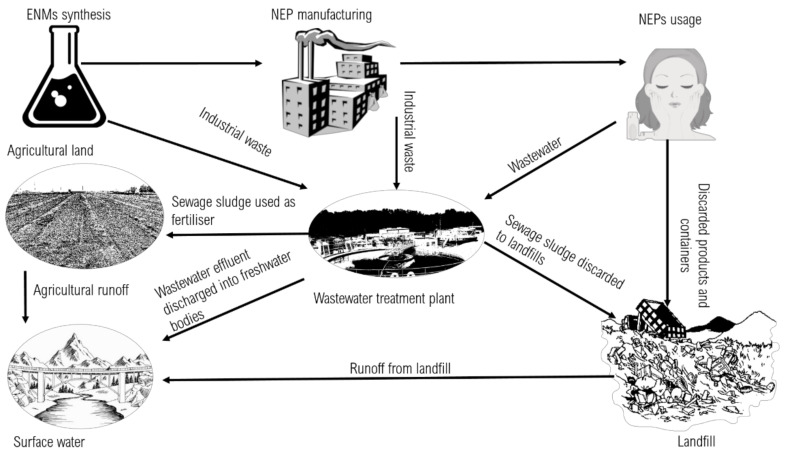
PR–ENMs pathway in the environment. The ENMs are depicted from manufacturing to different environmental exposure pathways in the environment.

**Figure 2 nanomaterials-11-02868-f002:**
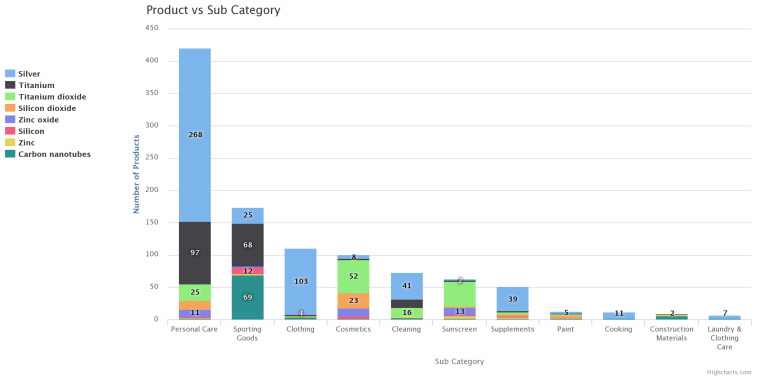
The incorporation of nAg, nTiO_2_, nZnO and nSiO_2_ in various nano-enabled products’ (NEPs) categories [53].

**Table 1 nanomaterials-11-02868-t001:** The global production and application of selected engineered nanomaterials (ENMs) based on beneficial properties.

ENMs Type	Global Production (Tons/Year)	References
nTiO_2_	10,000–15,000	[50,51]
nZnO	1000–36,000	[39,51]
nAg	420	[51]
nSiO_2_	1,400,000	[51]

**Table 3 nanomaterials-11-02868-t003:** The summary of reported on ecotoxicological data based on PR–ENMs ^1^ investigated, surrogate organisms observed effect and LC_50_
^2^ or EC_50_
^3^ where reported (*continuation*).

NEPs Type	ENMs Type	Organism	LC_50_ or EC_50_	Observation	References
Sunscreen	nTiO_2_ + nZnO	*P. lividus*	14–96 µL/L (standard salinity)	Growth inhibition	[148]
*P. tricornutum*	9.9–82 µL/L (salinity stress)
*C. orientalis*	
Textile	nAg	*D. rerio*	0.26–0.4 mg/L	Hatching inhibition	[147]
*	Abnormal embryo development
No effects	[124]
Household detergent	nAg	*P. aliena*	*	Growth inhibition	[122]
*C. fuciola*
*A. agilis*
*S. koyangensis*
Paint	nAg	*P. subcapitata*	*	No effects	[121]
*V. fischeri*
*D. magna*

^1^ Product released engineered nanomaterials; ^2^ Median lethal concentration, the concentration of a chemical (nanomaterials) that will kill 50 percent of the sample population being investigated; ^3^ Effective median concentration, the concentration of a substance in an environmental medium expected to produce an effect in 50% of test organisms in a given population; * EC_50_ or LC_50_ not reported.

**Table 4 nanomaterials-11-02868-t004:** The PR–ENMs risk characterisation estimates for different organisms.

NEPs Type	Organisms Group	PR–ENMs	RQ (Least Case Scenarios)	RQ Interpretation	RQ (Worst Case Scenarios)	RQ Interpretation
Sunscreen	Algae	PR–nTiO_2_	0.073	No significant risk	34.1	Potential adverse effects
		PR–nZnO	35.4	Potential adverse effects	35.4	Potential adverse effects
	Echinoderms	PR–nTiO_2_	0.007	No significant risk	0.083	No significant risk
		PR–nZnO	6.04	Small adverse effects	41.4	Potential adverse effects
	Crustacea	PR–nZnO	25.9	Potential adverse effects	25.9	Potential adverse effects
Textile	Fish	PR–nAg	800	Significant adverse effects	148,076.9	Significant adverse effects

## Data Availability

All data sources are cited, and a citation list is provided.

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
