# Peer review of "Aquatic Environment Exposure and Toxicity of Engineered Nanomaterials Released from Nano-Enabled Products: Current Status and Data Needs"

_nanomaterials, 2021, doi:10.3390/nano11112868_

Round 1

Reviewer 1 Report

In this manuscript, the authors reviewed the state of the art concerning the characterisation of release and toxicity effects of PR–ENMs in aquatic environments and estimated risk. The review identified key  gaps and outlines priority research needs on the environmental exposure and hazardous effects of PR–ENMs in water systems. The current review is the first to consolidate the release and effects data for risk estimation. In overall, this manuscript is interesting but in order to consider publication, this work should be revised. The following comments should be addressed for the improvement of their manuscript.

Comment 1: The overall study aims for this review study need to be further clarified in detail, especially on particle size range, crystallinity and content limitation of ENMs towards toxicity effects of PR–ENMs in aquatic environments and estimated risk.

Comment 2: The physical properties, chemical properties, FDA safety concerns and regulation on those selected ENMs should be further discussed and clarified in detail for better understanding purpose.

Comment 3: The future direction, and perspectives for ENMs as well as their toxicity effects and risks as compared to the non-nano sized materials need to be discussed in detail before the conclusion section.

Comment 4: The carefully English correction is necessary for the whole manuscript. There are several grammatical mistakes and errors were found throughout the manuscript. Please check and revise accordingly.

Author Response

Reviewer 1

Major suggestions

Response

The overall study aims for this review study need to be further clarified in detail, especially on particle size range, crystallinity and content limitation of ENMs towards toxicity effects of PR–ENMs in aquatic environments and estimated risk.

The authors couldn’t fully understand this review comment, especially concerning “particle size range, crystallinity”, however the Aim section was refined to improve clarity (Line 96 - 100). The content limitation is provided on the study criteria in the paragraph that follows (Line 108 - 114)

The physical properties, chemical properties, FDA safety concerns and regulation on those selected ENMs should be further discussed and clarified in detail for better understanding purpose.

Accepted. Line 67 - 72 have been added

The future direction, and perspectives for ENMs as well as their toxicity effects and risks as compared to the non-nano sized materials need to be discussed in detail before the conclusion section.

Accepted: the future direction is provided concerning country/region specific inventories, limiting assessments to priority NEPs with regards to environmental exposure potential for ENMs, enhancement of environmental realism, and hotspots prioritization; Line 641 - 658

However, the authors couldn’t understand the comment pertaining to non-nanosized materials as the review was focusing on nano sized materials. 

The carefully English correction is necessary for the whole manuscript. There are several grammatical mistakes and errors were found throughout the manuscript. Please check and revise accordingly.

Accepted, further grammatical corrections made.

Reviewer 2 Report

This manuscript, reviewing the impact in the aquatic environment of engineered nanomaterials released from commercial products is both interesting and timely. The manuscript was quite pleasant to read.

The review incorporates the primary literature from this area of research and has been competently integrated to provide a concise overview of the field. The manuscript is well-structured and the cited literature is appropriate for the subject matter.

The discussion appropriately compares and contrasts data gathered over the past number of years, and the conclusions are convincing. The authors have been very quantitative in listing the concentrations and sizes of nanomaterials which is particularly useful to the reader. The manuscript provides a convenient repository of the current state of knowledge in this area, and I would recommend publication in its current form.

Minor comments:

Line 352, Equation 1 – is there a typographical error for the numerator, should it be MEC and not PEC?

Please check the english as there are spelling/typo errors and syntax in places is problematic.

Author Response

Reviewer 2

Major suggestions

Response

Line 352, Equation 1 – is there a typographical error for the numerator, should it be MEC and not PEC?

Accepted: correction made in Line 575

Please check the english as there are spelling/typo errors and syntax in places is problematic.

Accepted, further grammatical corrections made.

Reviewer 3 Report

This is an important contribution and a very good comprehensive review.  I commend the authors for this work.  I found only a few minor issues and I also provide some suggestions to make the review more impactful

Suggestions

Lines 53-54 makes a very important statement.  This flows throughout the review.  It would be nice to have this as a separate paragraph in the introduction and emphasize more as this is a critical issue.  Perhaps comments on the timelines of release as a product to their use.  Ie most Ag NP textiles have only came out since about 2009.  Although exponential use it is only really about 5 yrs since widespread use, so should we be surprise there is little data.  Same arguments around NPs in cosmetics.

Line 70;  I agree with the focus on articles from 2010.  But is there any key ground-breaking or originating work pre 2010 that could still be mentioned in a brief historical perspective

I think it would be valuable to have a short paragraph pointing out what analytical methods are presently used and their challenges.  One big one I know of is separation challenges due to environmental samples, particularly WWTP is full of natural nanomaterials and metal ions.

The paper could benefit from a cartoon to display the environmental pathway.  Ie NM manufacturing (this has toxic waste), NEP manufacturing (this would have waste), use NEPs, release (general environment vs WWTP) subsequent release to environment. One can simply indicate certain toxic streams are not being evaluated.

             Related to this it would be good to point out that Waste water treatment plants (WWTP) are the gateway between the PR-EMN to the environment.  Given that about 50% of the ENM are trapped in the solid waste sludge and that this sludge is often converted to fertilizer, this leads to possibilities to introduced into aquatic systems via run off.  I think this is also worthy of a paragraph in combination with the cartoon.

Although indirectly discussed (line 340).  Perhaps a separate paragraph to highlight more possible issues of difference between acute vs chronic.  Thoughts could also include bioaccumulation in the periphyton and plants.

Although beyond this review, some comment about the cap or coating of nanoparticles.  In synthesis they are there for properties of size and stability but can also contribute to their application properties but of course toxicity.  I know there is next to nothing looking at this but it is important to mention in your review for completion.

Minor corrections

Tables:  I personally don’t like abbreviations used in Titles and Tables for headers etc.   At the very least one should footnote the definitions.  This is because many people look at figures and tables before they read the text.

Line 19 and again line 51.  I suggested to define product released separated ie ‘product released (PR) engineering nanomaterials (PR_ENM)’  as PR is used on its own elsewhere.

Line 62; reviews’ might be better than reviewed

Equation 1 line352;  PEC I think you mean MEC

I could not see where LC50 and EC50 are defined.   Also it is not clear in Table 3 and in text if you mean CLC50 divided by EC50 or is it LC50 or EC50.  This also comes up in your equation 2 line 356.  Also you have indicated this in a different order  as EC50/LC50, should be consistent

Line 291, Zn2+  should be Zn2+ 

My reference list ended at reference 72 so I was unable to validate if key references were used, but I am sure they caught most.

Author Response

Reviewer 3

Major suggestions

Response

Lines 53-54 makes a very important statement.  This flows throughout the review.  It would be nice to have this as a separate paragraph in the introduction and emphasize more as this is a critical issue.  Perhaps comments on the timelines of release as a product to their use.  Ie most Ag NP textiles have only came out since about 2009.  Although exponential use it is only really about 5 yrs since widespread use, so should we be surprise there is little data.  Same arguments around NPs in cosmetics.

Accepted: A paragraph specifically expanding the statement has been, Lines 56 - 72

The latter section (“Perhaps….arguments”) could not be understood by the authors, however all the reviewed studies given in Tables were dated.

Line 70; I agree with the focus on articles from 2010.  But is there any key ground-breaking or originating work pre 2010 that could still be mentioned in a brief historical perspective

Accepted: some of the ground-breaking works were published pre-2010 and have now been included herein. The 3 studies [15, 16, 17] that could be obtained pre-2010 have been added under the release section “Section 3”. Lines 312 – 317 and 367 – 378. Furthermore, lowering of search criteria to 2008 only resulted in the inclusion of the three papers mentioned. 

I think it would be valuable to have a short paragraph pointing out what analytical methods are presently used and their challenges.  One big one I know of is separation challenges due to environmental samples, particularly WWTP is full of natural nanomaterials and metal ions.

Accepted: rather we have provided a handful of detailed recent reviews on this aspect which a reader can consult for a detailed account. Line 81

The paper could benefit from a cartoon to display the environmental pathway.  Ie NM manufacturing (this has toxic waste), NEP manufacturing (this would have waste), use NEPs, release (general environment vs WWTP) subsequent release to environment. One can simply indicate certain toxic streams are not being evaluated.

Accepted: A schematic diagram (Figure 1) has been added to illustrate the environmental pathway of ENMs from various sources. Line 91

Related to this it would be good to point out that Waste water treatment plants (WWTP) are the gateway between the PR-EMN to the environment.  Given that about 50% of the ENM are trapped in the solid waste sludge and that this sludge is often converted to fertilizer, this leads to possibilities to introduced into aquatic systems via run off.  I think this is also worthy of a paragraph in combination with the cartoon.

Accepted: We have added a paragraph; Line 83 - 90 to highlight the relevance of WWTPs as sinks and sources of ENMs to water resources.

Although indirectly discussed (line 340).  Perhaps a separate paragraph to highlight more possible issues of difference between acute vs chronic.  Thoughts could also include bioaccumulation in the periphyton and plants

Accepted: we have expanded the account in Line 561 – 562 to emphasize the influence of exposure prolongation (acute vs chronic) in aquatic systems. However, it noteworthy that no chronic studies using PR-ENMs were accessible and hence it is a gap that needs to be filled.

Although beyond this review, some comment about the cap or coating of nanoparticles.  In synthesis they are there for properties of size and stability but can also contribute to their application properties but of course toxicity.  I know there is next to nothing looking at this but it is important to mention in your review for completion.

Not accepted: the authors agree with the Reviewer with the regards to the coating influence and toxicity of ENMs. However, the influence of coating has been examined in studies with pristine engineered nanomaterials (for instance: http://doi.org/10.1111/cpr.12192 ; https://pubs.acs.org/doi/10.1021/es301977w). With regards to NEPs where coating has been characterised (e.g.: https://doi.org/10.3390/nano11102537 ); such studies only characterize extracted and released ENMs but do not assess the toxicity. Hence the aspect of ENMs capping could not be incorporated in the current review due to scope misalignment.

Minor suggestions

Response

Tables:  I personally don’t like abbreviations used in Titles and Tables for headers etc.   At the very least one should footnote the definitions.  This is because many people look at figures and tables before they read the text.

Accepted: changes made in Tables 2-5

Line 19 and again line 51.  I suggested to define product released separated ie ‘product released (PR) engineering nanomaterials (PR_ENM)’  as PR is used on its own elsewhere.

Accepted and edited accordingly, Lines 17 and 53

Line 62; reviews’ might be better than reviewed

Accepted and edited accordingly, Line 95

Equation 1 line352;  PEC I think you mean MEC

Corrected, Line 575.

I could not see where LC50 and EC50 are defined.   Also it is not clear in Table 3 and in text if you mean CLC50 divided by EC50 or is it LC50 or EC50.  This also comes up in your equation 2 line 356.  Also you have indicated this in a different order  as EC50/LC50, should be consistent

EC50/LC50 re-written as EC50 or LC50. Throughout text.

Equations separated for clarity, Line 579

EC50/LC50 defined below Table 3, Line 547 - 549.

Line 291, Zn2+  should be Zn2+ 

Accepted, fixed Line 481

Round 2

Reviewer 1 Report

In overall, this manuscript was technically well revised. This revised manuscript meets the criteria of nanomaterials. Therefore, in my opinion, the revised manuscript can be accepted for publication.